# Lipocalin 2 Deficiency Alters Prostaglandin Biosynthesis and mTOR Signaling Regulation of Thermogenesis and Lipid Metabolism in Adipocytes

**DOI:** 10.3390/cells11091535

**Published:** 2022-05-03

**Authors:** Jessica Deis, Te-Yueh Lin, Theresa Bushman, Xiaoli Chen

**Affiliations:** Department of Food Science and Nutrition, University of Minnesota, Twin Cities, MN 55455, USA; jessicadeis@gmail.com (J.D.); linx0916@umn.edu (T.-Y.L.); bushm024@umn.edu (T.B.)

**Keywords:** lipocalin 2, prostaglandin, mTOR signaling, adipocyte

## Abstract

Apart from a well-known role in the innate immune system, lipocalin 2 (Lcn2) has been recently characterized as a critical regulator of thermogenesis and lipid metabolism. However, the physiological mechanism through which Lcn2 regulates cellular metabolism and thermogenesis in adipocytes remains unknown. We found that Lcn2 expression and secretion are significantly upregulated by arachidonic acid (AA) and mTORC1 inhibition in differentiated inguinal adipocytes. AA-induced Lcn2 expression and secretion correlate with the inflammatory NFkB activation. Lcn2 deficiency leads to the upregulation of cyclooxygenase-2 (COX2) expression, as well as increased biosynthesis and secretion of prostaglandins (PGs), particularly PGE2 and PGD2, induced by AA in adipocytes. Furthermore, Lcn2 deficiency affects the mTOR signaling regulation of thermogenic gene expression, lipogenesis, and lipolysis. The loss of Lcn2 dismisses the effect of mTORC1 inhibition by rapamycin on COX2, thermogenesis genes, lipogenesis, and lipolysis, but has no impact on p70 S6Kinase-ULK1 activation in Lcn2-deficient adipocytes. We conclude that Lcn2 converges the COX2-PGE2 and mTOR signaling pathways in the regulation of thermogenesis and lipid metabolism in adipocytes.

## 1. Introduction

Lipocalin 2 (Lcn2), or neutrophil gelatinase-associated lipocalin, is a 25 kD secreted protein first characterized in neutrophils [1]. It belongs to the lipocalin supergene family of hydrophobic ligand binding proteins [2,3] and plays an important role in innate immune responses [4,5]. In addition to neutrophils and other immune cells, Lcn2 is also expressed in a range of tissues including the liver, bone marrow, uterus, prostate, stomach, lung, colon, and adipose tissue [6]. The promoter region of the Lcn2 gene contains binding motifs for nuclear factor kappa-light-chain-enhancer of activated B cells (NFκB) and CCAAT enhancer binding protein (C/EBP), as well as several nuclear receptor response elements including the glucocorticoid response element, estrogen response element, and retinoic acid receptor response element [7,8,9]. In adipocytes, Lcn2 expression and secretion are abundantly induced by metabolic stress, inflammatory stimulation, as well as nutrients such as high glucose and a variety of lipids [10]. Together, these features suggest that Lcn2 has a diversity of functions in the cell.

Lcn2 was originally found to bind to and stabilize matrix-metalloproteinase-9 (MMP-9), which is involved in extracellular matrix remodeling [1,11]. Later, the activation of toll-like receptors (TLRs) on the surface of immune cells by lipopolysaccharide (LPS) was found to enhance the transcription and secretion of Lcn2 [5,12]. This led to a well-characterized role in the innate immune system, where it has been shown to inhibit bacterial growth by sequestering iron-containing siderophores [5,13]. As such, Lcn2-deficient mice are more susceptible to bacterial infections [4,5] and high-fat-diet-induced gut microbiota dysbiosis [14]. In this way, Lcn2 is upregulated by inflammatory stimuli but plays a protective role in the body’s response to infection. Moreover, recent investigations indicate that Lcn2 also plays a role in the regulation of energy metabolism and adipose tissue inflammation in obesity and diabetes [15,16,17]. For instance, our studies have shown that Lcn2 expression is critical for brown adipose tissue (BAT) thermogenesis and beiging of inguinal white adipose tissue (Ing-WAT) [16,17,18]. However, how Lcn2 regulates cellular metabolism and thermogenesis in response to metabolic stress and nutrients in adipocytes remains unknown.

Prostaglandins have been implicated in adipogenesis, beiging of white adipocytes, and adipose tissue inflammation in obesity and insulin resistance [19,20]. Prostaglandins are derived from arachidonic acid (AA), which is cleaved from membrane phospholipids by phospholipases A2 (PLA2). *Cyclooxygenase*-1 (COX-1) and COX-2 synthesize prostaglandin H_2_ (PGH2) from arachidonic acid (AA), which can then be further metabolized to prostaglandin E_2_ (PGE2), prostaglandin D_2_ (PGD2), prostaglandin J_2_ (PGJ2), and thromboxane. COX-1 is thought to be constituently active, whereas COX-2 is activated in response to inflammatory stimuli. Therefore, perturbations in COX-2 levels are more suggestive of changes in prostaglandin synthesis induced by inflammation. For instance, levels of multiple PGs and COX2 mRNA expression are increased in adipose tissue of obese mice [21].

Mammalian/mechanistic target of rapamycin (mTOR) has been considered as a cellular metabolic hub that integrates multiple signaling cascades. The cooperation of COX-2 and mTOR is known to act on regulating cellular metabolism, including thermogenesis and lipid metabolism. Therefore, it is of importance to understand whether and how PG and mTOR signaling contributes to the metabolic regulation of Lcn2 in adipocytes. The objective of this study is to investigate the impact of Lcn2 deficiency on PG synthesis and mTOR signaling, which connects PG signaling to thermogenesis in beige adipocytes. We found that the levels of PGs, particularly PGE2 and COX2 expression, were significantly upregulated in Lcn2-deficient adipocytes. Lcn2 deficiency attenuates the mTOR signaling regulation of thermogenic gene expression, lipogenesis, and lipolysis. Our data suggest that Lcn2 participates in the network of COX2-PGs-mTOR in the regulation of thermogenesis and lipid metabolism in adipocytes.

## 2. Materials and Methods

### 2.1. Animals

Lcn2-deficient mice were kindly provided by Dr. Alan Aderem, Institute for Systems Biology, Seattle, Washington, USA. A heterozygous mating scheme was used to generate wild-type (WT) and Lcn2^−/−^ mice as previously described [15]. Animals were housed at 22 °C in a specific pathogen-free facility at the University of Minnesota. Animal studies were conducted with the approval of the University of Minnesota Animal Care and Use Committee and conformed to the National Institute of Health guidelines for laboratory animal care.

### 2.2. Cell Culture and Differentiation of Primary Stromal-Vascular Cells

Stromal-vascular (SV) cells were isolated from inguinal white adipose tissue (WAT) of WT and Lcn2^−/−^ male mice as previously described [16]. Cells were grown to confluence in Dulbecco’s Modified Eagle’s Medium (DMEM, Invitrogen, Grand Island, NY, USA) containing L-glutamine, 25 mM glucose, 10% FBS (Atlanta Biological, Lawrenceville, GA, USA), and 100 units/mL of penicillin/streptomycin (Invitrogen, Carlsbad, CA, USA). After reaching confluence, SV cells were treated for 2 days with a differentiation cocktail containing DMEM, 10% FBS, penicillin/streptomycin, 0.25 mM isobutylmethylxanthine (Sigma, St. Louis, MO, USA), 0.5 µM dexamethasone (Sigma, St. Louis, MO, USA), and 0.85 µM insulin. The cells were then maintained in DMEM with 10% FBS and 0.85 µM insulin until fully differentiated. Fully differentiated adipocytes were serum-starved in DMEM containing 0.5% FBS for 3–6 h prior to addition of insulin and various treatments, as indicated in figure legends.

### 2.3. Measurement of Prostaglandins

Liquid chromatography–tandem mass spectrometry (LC–MS/MS) was used to quantify eicosanoid secretion from adipocytes. Differentiated inguinal adipocytes from WT and LCN2^−/−^ mice were treated without or with 150 nM arachidonic acid (AA) for 3 h, 24 h, and 48 h, and cell culture media were collected for the measurement of prostaglandins. For whole media analysis, deuterated standard containing a known mix of eicosanoids was added to the sample upon collection. The eicosanoids in the media samples were collected by running them through a reversed phase STRATA-X column. The eicosanoids retained on the column were eluted in 1 mL of HPLC-grade methanol containing 2% formic acid. After elution, the samples were dried under nitrogen, flushed with nitrogen, and immediately capped. The samples were analyzed by LC–MS/MS at the Mass Spectrometry and Proteomics Center at the University of Minnesota using a previously validated protocol for eicosanoid quantification [22].

### 2.4. Quantitative Real-Time PCR

Total RNA was isolated from cells using TRIZOL reagent (Invitro, Carlsbad, CA, USA). RNA was DNAase-treated prior to the synthesis of cDNA using Superscript II reverse transcription kit (Invitrogen, Carlsbad, CA, USA). Real-time quantitative PCR was conducted using the SYBR Green qPCR Master Mix (SABiosciences, Frederick, MD, USA) with a StepOne Real-time PCR System (Applied Biosystem, Foster City, CA, USA). The ΔΔCt method was used to calculate mRNA expression and *Actin* served as an internal control.

### 2.5. Western Blot Analysis

Equivalent amounts of protein were run on an SDS–PAGE gel and transferred to a nitrocellulose membrane prior to incubation with primary and secondary antibodies. The sources of primary antibodies were as follows: Lcn2 and UCP1 (R&D systems, Minneapolis, MN, USA); NFkB, phosphorylated NFkB, COX2, p70 S6Kinase, phosphorylated p70 S6Kinas, and β-actin (Cell Signaling Technology, Danvers, MA, USA); ULK1 and phosphorylated ULK1 (555 and 757) (Santa Cruz Biotech, Dallas, TX, USA). Secondary antibodies were from R&D Systems (Minneapolis, MN, USA). ECL Western blotting substrate (Pierce, Rockford, IL, USA) was used to detect reactivity.

### 2.6. Glycerol Assay

Differentiated inguinal adipocytes were cultured in fresh KRB containing 2% fatty acid-free BSA, 0.1% glucose, and 0.85 µM insulin with or without 10 µM isoproterenol (Iso) in the presence or absence of 25 nM Rapamycin. The culture medium was collected after 3 h of incubation and glycerol release was measured in aliquots from incubation buffer using free glycerol reagent (Sigma), following the instructions provided by the manufacturer.

### 2.7. Statistical Analysis

Values are reported as mean ± standard error of the mean (SEM). Statistical significance was determined by the two-tailed Student’s *t* test, where a *p*-value less than 0.05 was considered significant.

## 3. Results

### 3.1. Lcn2 Regulates AA-Induced Prostaglandin Biosynthesis and Secretion in Beige Adipocytes

AA induces Lcn2 expression and secretion. Arachidonic acid (AA) is the precursor of PG and leukotrienes. First, we determined if AA regulates Lcn2 expression and secretion when added to the culture media of inguinal SV-differentiated adipocytes. After 6 h of AA treatment, only very low levels of secreted Lcn2 were detected in culture media, but cellular Lcn2 was undetectable (Figure 1A). However, Lcn2 levels particularly in the culture media were dramatically increased for 24 h upon AA treatment (Figure 1B). This time-dependent induction of Lcn2 by AA was correlated with the upregulation of NFkB phosphorylation in WT inguinal adipocytes (Figure 1). These data suggest that AA induces Lcn2 expression and secretion as a result of inflammatory NFkB activation.

Lcn2 deficiency alters AA-induced PG biosynthesis and secretion in adipocytes. To determine the effect of Lcn2 deficiency on PG biosynthesis, WT and Lcn2 KO differentiated inguinal adipocytes were treated with or without 150 nM arachidonic acid in the presence of insulin for 48 h. The conditioned media were collected for the measurement of PGs using liquid chromatography–tandem mass spectrometry (LC–MS/MS). As shown in Figure 2A, the addition of AA for 48 h significantly increased the PGE2 secretion in WT adipocytes and at a much higher level in Lcn2 KO adipocytes. As the levels of PGE2 in 48 h AA-treated culture media were relatively low, we sought to determine the effect of a short period of AA treatment on PG biosynthesis. Cells were treated with or without AA for 3 h and 24 h, respectively. After AA treatment for 3 h, there were no significant differences in the biosynthesis and secretion of all the PGs that we detected between WT and Lcn2 KO adipocytes (Figure 2B–G). Upon AA treatment for 24 h, the levels of secreted PGE2 and PGD2 were significantly increased in Lcn2 KO adipocytes compared to WT adipocytes (Figure 2B,G). While the levels of 6-keto PGF1α and PGF2α had a trend toward an increase in Lcn2 KO adipocytes, there was no difference in 15d-PGJ2 and AA between WT and KO cells with the 24 h AA treatment (Figure 2F). Further, we examined the expression of genes encoding the enzymes involved in the biosynthesis of PGs and leukotrienes in adipocytes. The results showed that the gene expression of *Cox2*, *Lta4h*, and *Ltc4s* was upregulated, whereas the expression of *Pla* and *Ptges2* was downregulated in Lcn2 KO adipocytes compared to WT cells (Figure 2H). This indicates that Lcn2 deficiency disrupts the PG biosynthesis and metabolism in adipocytes.

### 3.2. Lcn2 Mediates the Regulation of mTOR Signaling in Prostaglandin Biosynthesis

mTORC1 inhibition induces Lcn2 expression and secretion in the presence of insulin in a time- and dose-dependent manner. It has been known that mTOR is one of the downstream targets of the COX2-PGE2 pathway [23,24]. The mTOR signaling pathway is known to play a role in the regulation of thermogenesis and beiging of adipose tissue [25,26,27,28]. It is of interest to determine if Lcn2 has a connection with the COX2-PGE2-mTOR pathway in the context of thermogenesis and lipid metabolism. First, we examined the regulation of Lcn2 expression and secretion by mTOR inhibition to determine the connection of Lcn2 to the mTOR signaling pathway. As illustrated in Figure 3A, Lcn2 mRNA expression was significantly upregulated by rapamycin, an mTORC1 inhibitor, in a dose-dependent manner, and the presence of insulin is required for the rapamycin induction. Likewise, Lcn2 protein secretion was induced by rapamycin in a time- and dose-dependent manner (Figure 3B,C). While 6 h of rapamycin treatment had no effect on Lcn2 secretion, 24 and 48 h of treatment increased the secretion of Lcn2 protein (Figure 3B). Additionally, we showed the dose-dependent effect of rapamycin on Lcn2 secretion (Figure 3C). When treated for 24 h, only 25 nM but not 2.5 nM rapamycin was able to stimulate the secretion of Lcn2. In both time- and dose-dependent studies, the effect of rapamycin requires the presence of insulin.

Lcn2 deficiency diminishes the inhibitory effect of rapamycin on the COX2 pathway. Next, we determined how Lcn2 deficiency affects the inhibitory effect of rapamycin on COX2. As shown in Figure 4A, rapamycin treatment for 24 h caused a reduction in protein levels of COX2 in WT adipocytes. However, Lcn2 deficiency attenuated this effect of rapamycin (Figure 4A). To determine if Lcn2 deficiency affects mTOR signaling, we examined the p70 S6Kinase and ULK1 activation by mTOR inhibition in Lcn2 KO adipocytes. The results indicate that the inhibition of mTORC1 by rapamycin and mTORC2 by Torin was not affected by Lcn2 deficiency. The p70 S6Kinase phosphorylation was similarly reduced by rapamycin in both WT and Lcn2 KO adipocytes (Figure 4B). Moreover, the phosphorylation of Unc-51 such as autophagy activating kinase (ULK1), a downstream target of mTOR signaling activation and a component of the autophagy pathway, was downregulated by both rapamycin and Torin to a similar extent in WT and Lcn2 KO adipocytes (Figure 4C). These results suggest that Lcn2 deficiency affects the COX2-PGE2 through an mTOR downstream mechanism.

### 3.3. Lcn2 Deficiency Attenuates the Effect of mTORC1 Inhibition on Thermogenesis, Lipogenesis, and Lipolysis

Lcn2 deficiency attenuates the effect of mTORC1 inhibition on thermogenesis. Both COX2 and mTOR have been known to play a role in thermogenesis and beiging of WAT [29,30,31]. We sought to understand how Lcn2 deficiency affects the effect of mTOR signaling on thermogenesis. Consistent with our previously published studies [18], *Pgc1α* gene expression was significantly decreased in Lcn2 KO inguinal adipocytes in the basal condition (Figure 5A). Rapamycin treatment for 24 h in the presence of insulin caused a significant downregulation of *Pgc1α* gene expression in WT adipocytes (Figure 5A). However, the inhibitory effect of rapamycin on *Pgc1α* was attenuated in Lcn2 KO adipocytes. Although they did not reach a statistical significance level, the changes in *Ucp1* gene (Figure 5B) and UCP1 protein expression (Figure 5C) followed a similar trend in WT and Lcn2 KO adipocytes treated with rapamycin treatment for 24 h. Rapamycin treatment had less effect on the downregulation of *Ucp1* gene expression in Lcn2 KO adipocytes compared to WT adipocytes (Figure 5B). The UCP1 protein expression was reduced by rapamycin treatment in the basal and Iso-treated conditions in WT adipocytes but not in Lcn2 KO adipocytes (Figure 5C). However, quantitative band intensities of Western blotting results from two experiments indicate that the differences between control and rapamycin-treated groups did not reach a statistical significance level in either WT or Lcn2 KO adipocytes (data not shown).

Lcn2 deficiency attenuates the effect of mTORC1 inhibition on lipogenesis. We then examined the expression of genes involved in lipogenesis in WT and Lcn2 KO adipocytes treated with or without 25 nM rapamycin in the presence of insulin. After 24 h of treatment, rapamycin significantly increased the gene expression of *Fasn*, *Srebp2*, and *Pepck* in WT adipocytes (Figure 6A–C). In Lcn2 KO adipocytes, the induction of *Fasn*, *Srebp2*, and *Pepck* gene expression by rapamycin was attenuated (Figure 6A–C); the expression levels of *Fasn*, *Srebp2*, and *Pepck* genes by rapamycin were significantly lower compared to WT adipocytes (Figure 6A–C). Although the expression of *Pparγ* was induced by rapamycin in WT and Lcn2 KO adipocytes to a similar extent, the level of *Pparγ* gene expression was significantly lower in Lcn2 KO adipocytes compared to WT cells under the basal and rapamycin-treated conditions (Figure 6D).

Lcn2 deficiency attenuates the effect of mTORC1 inhibition on lipolysis. A similar impact of Lcn2 deficiency on lipolysis was also observed in adipocytes treated with rapamycin. As shown in Figure 7A,B, 24 h of treatment of rapamycin upregulated the expression of *Hsl* and *Atgl* genes in WT adipocytes but not in Lcn2 KO adipocytes. Further, we conducted a lipolytic assay and treated WT and Lcn2 KO adipocytes with or without rapamycin in the presence or absence of isopropanol for 3 h. Conditioned media were collected for the glycerol assay. The results demonstrated that both basal and Iso-stimulated levels of glycerol were lower in Lcn2 KO adipocytes compared to WT cells (Figure 7C). Interestingly, the levels of glycerol release induced by the treatment with the combination of rapamycin and isopropanol (Iso) were significantly lower in Lcn2 KO adipocytes (Figure 7C). Together with the lipolytic gene expression, these data suggest that Lcn2 deficiency attenuates the effect of mTORC1 inhibition on lipolysis in adipocytes.

## 4. Discussion

The role of both COX2-PGE2 and mTOR signaling pathways in the regulation of thermogenesis, lipid metabolism, and inflammation in adipose tissue has been extensively studied. Several studies have implicated the connection of these two pathways in terms of their role in cellular metabolism [23,24]. In this study, we showed that Lcn2 plays a role in thermogenesis and lipid metabolism involving the COX2-PGE2 and mTOR signaling pathways. We found that arachidonic acid (AA) as a precursor of PG synthesis dramatically induces Lcn2 secretion, and Lcn2 deficiency leads to increased PGE2 synthesis and secretion in differentiated inguinal adipocytes. Lcn2 expression and secretion are also upregulated by inhibiting the mTOR signaling pathway. Lcn2 deficiency attenuates the effect of mTORC1 inhibition on thermogenesis, lipogenesis, and lipolysis in inguinal adipocytes.

Arachidonic acid is oxygenated and further converted into a variety of PG and leukotriene products known as inflammatory mediators [32]. To determine how Lcn2 links to the production of AA-induced PGs, we examined the AA regulation of Lcn2 expression and secretion in adipocytes. Our results demonstrated that Lcn2 expression and secretion are upregulated by AA in a time-dependent manner. The AA induction of Lcn2 correlates with AA-induced NFkB activation. For example, the significant induction of both Lcn2 secretion and NFkB phosphorylation was observed upon 24 h but not 6 h of AA stimulation. It is likely that AA treatment induces Lcn2 expression via activating the NFkB inflammatory pathway. This possibility is supported by the fact that the Lcn2 gene promoter region contains an NFkB binding site, as well as our previously published data showing that blocking NFkB activation could significantly reduce inflammation-induced Lcn2 expression and secretion [10]. Additionally, it is also possible that AA induces Lcn2 expression indirectly through stimulating PG production, which, in turn, activates NFkB and Lcn2 expression. Next, we found that Lcn2 deficiency alters AA-induced PG production in a time-dependent manner. For example, Lcn2 deficiency does not affect PGE2 levels in the culture media of adipocytes in the basal and 3 h AA-treated condition. However, increased PGE2 and PGD2 levels and COX2 expression were observed in Lcn2 KO adipocytes after 24 h of AA treatment. The results indicate that Lcn2 is involved in the regulatory network of PG biosynthesis likely as a negative feedback mechanism through inhibiting COX2 activity.

The mTOR plays a critical role in mediating the effect of metabolic stress and nutrients on thermogenesis and lipid metabolism. Studies have indicated that the mTORC1 pathway is required for the cAMP-PKA signaling regulation of thermogenesis in adipose tissue [26,27,33]. The activation of the cAMP-PKA pathway by cold or the β3-adrenergic receptor agonist (CL316,243) not only activates thermogenesis but also simultaneously stimulates de novo lipogenesis (DNL) in subcutaneous white adipose tissue as well as brown adipose tissue [34,35]. Moreover, lipolysis is required for the thermogenic activation and UCP1 upregulation in sWAT [35]. The inhibition of mTORC1 by rapamycin abolishes the CL316,243-stimulated upregulation of *Ucp1* but not DNL genes [36]. The mTORC1 inhibition has also been shown to stimulate lipolysis [37,38,39].

Several studies have linked the COX2-PGE2 to the mTOR signaling pathway; PGE2 has been known to activate the mTOR signaling pathway [23,24]. We then attempted to address the questions of (1) how Lcn2 regulates the role of mTOR signaling in thermogenesis and lipid metabolism and (2) whether Lcn2 connects COX2 to mTOR signaling. We found that the inhibition of mTOR by rapamycin induces Lcn2 gene expression and protein secretion in a time- and dose-dependent manner, and this induction requires the presence of insulin. Interestingly, Lcn2 deficiency diminishes the inhibitory effect of rapamycin on COX2 protein levels but does not seem to affect the rapamycin-induced reduction in the *Ptges2* expression. To understand how Lcn2 deficiency influences the inhibitory effect of rapamycin on the function of the mTOR signaling pathway, we examined the downstream targets of mTOR signaling such as p70 S6kinase and ULK1 activation. Our results showed that both rapamycin and Torin could inhibit the phosphorylation of p70 S6kinase and ULK1 effectively in both WT and Lcn2 KO adipocytes, suggesting that Lcn2 likely acts as a negative feedback mechanism on the downstream of p70S6kinase-ULK I in the mTOR signaling pathway to regulate the COX2-PGE2 production.

In terms of the role of Lcn2 in the regulation of mTOR signaling in thermogenesis and lipid metabolism, we found that Lcn2 deficiency attenuates the effect of rapamycin on the downregulation of *Pgc1α* gene expression, as well as *Ucp1* gene and protein expression. This suggests that Lcn2 is also part of the mTOR signaling in the regulation of thermogenesis in beige adipocytes. More interestingly, our results showed that mTORC1 inhibition by rapamycin leads to the significant upregulation of DNL genes such as *Fasn*, *Srebp2*, and *Pepck* in WT adipocytes. However, the rapamycin-induced upregulation of DNL genes was almost completely blunted in adipocytes lacking Lcn2. A similar effect of Lcn2 deficiency was also observed on rapamycin-induced lipolysis. For instance, rapamycin treatment significantly upregulates *Hsl* and *Atgl* gene expression in WT but not Lcn2 KO adipocytes. Furthermore, the glycerol assay of adipocyte-conditioned media indicates that Lcn2 KO adipocytes have lower levels of glycerol release in the basal and Iso-stimulated states. Rapamycin treatment enhances Iso-stimulated glycerol release in WT adipocytes, but no such effect was observed in Lcn2 KO adipocytes. These results suggest that Lcn2 deficiency disrupts the mTOR signaling regulation of lipid metabolism, specifically lipogenesis and lipolysis, in beige adipocytes.

In summary, we found that Lcn2 expression and secretion in inguinal adipocytes are upregulated by AA and mTORC1 inhibition. Lcn2 deficiency leads to the upregulation of COX2 and increased production of PGE2 and PGD2 induced by AA. Lcn2 deficiency also diminishes the effect of mTORC1 inhibition on COX2, thermogenesis genes, lipogenesis, and lipolysis. The data lead us to conclude that Lcn2 plays a role in the COX2-PG biosynthesis and mTOR signaling regulation of thermogenesis, lipogenesis, and lipolysis in adipocytes. How Lcn2 converges these two pathways in energy metabolism warrants further investigations.

## Figures and Tables

**Figure 1 cells-11-01535-f001:**
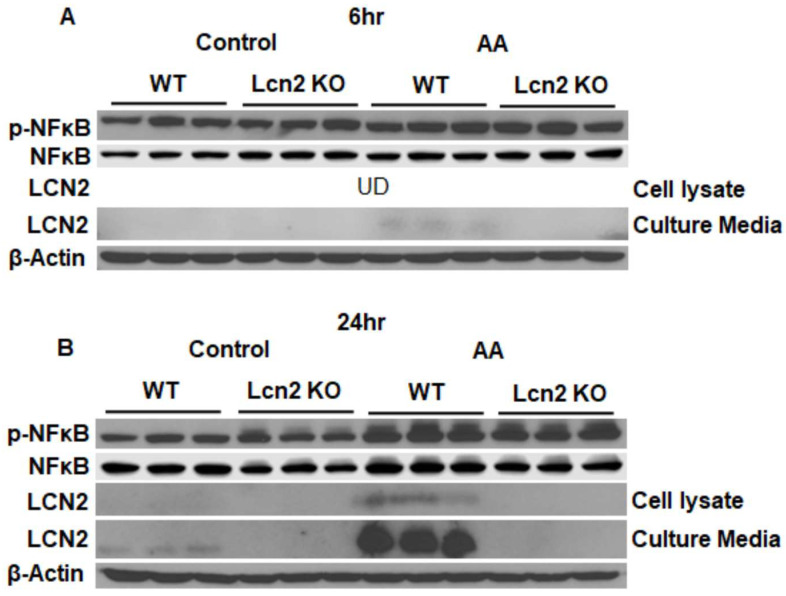
Regulation of Lcn2 expression and secretion by AA in differentiated inguinal adipocytes. Lcn2 expression and secretion and NFkB phosphorylation in inguinal adipocytes treated with 150 nM AA for 6 h (**A**) and 24 h (**B**). *n* = 3.

**Figure 2 cells-11-01535-f002:**
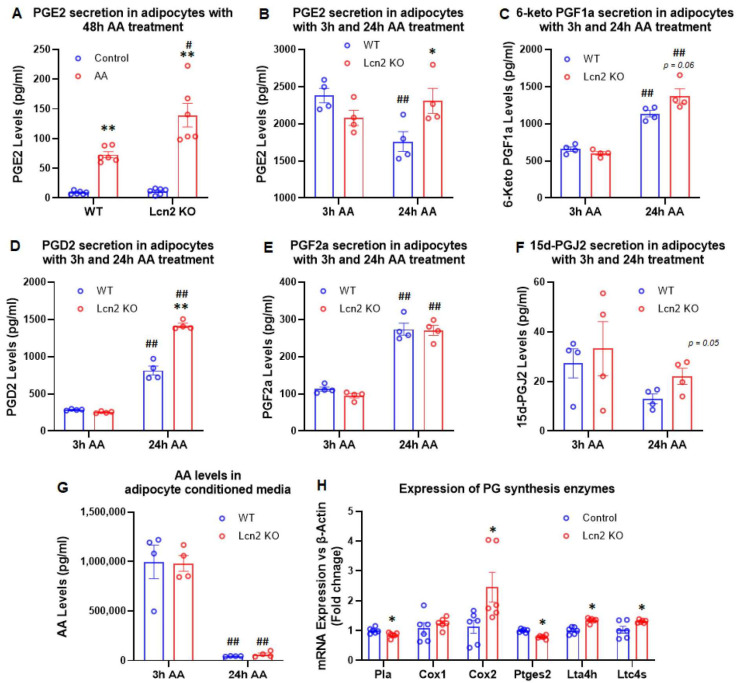
Prostaglandin production induced by AA in inguinal adipocytes. (**A**) PGE2 secretion in inguinal adipocytes treated with 150 nM AA for 48 h. (**B**–**F**) Secreted levels of PGE2, 6-keto PGF1a, PGD2, PGF2a, and 15d-PGJ2 in inguinal adipocytes treated with 150 nM AA for 3 h and 24 h. (**G**) AA levels in conditioned media of inguinal adipocytes. (**H**) Gene expression of PG synthesis enzymes in inguinal adipocytes. Results are presented as mean ± SEM (*n* = 4 for (**A**–**G**); *n* = 6 for (**H**)). * *p* < 0.05, ** *p* < 0.01 versus WT adipocytes. ^#^
*p* < 0.05, ^##^
*p* < 0.01 versus 3 h AA treatment.

**Figure 3 cells-11-01535-f003:**
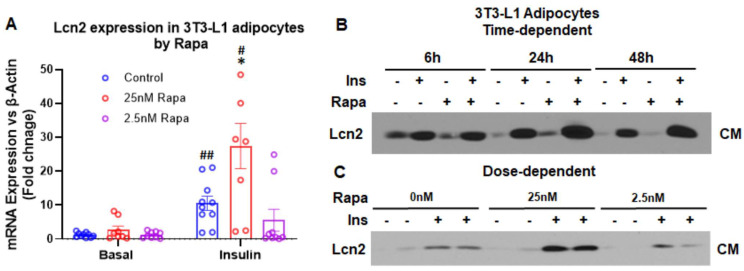
Lcn2 expression and secretion by rapamycin in 3T3-L1 adipocytes. (**A**) Dose-dependent mRNA expression of Lcn2. (**B**) Time- and dose-dependent secretion of Lcn2 protein by rapamycin. Results are presented as mean ± SEM (*n* = 6 for A). * *p* < 0.05 versus controls. ^#^
*p* < 0.05, ^##^
*p* < 0.01 versus corresponding basal group.

**Figure 4 cells-11-01535-f004:**
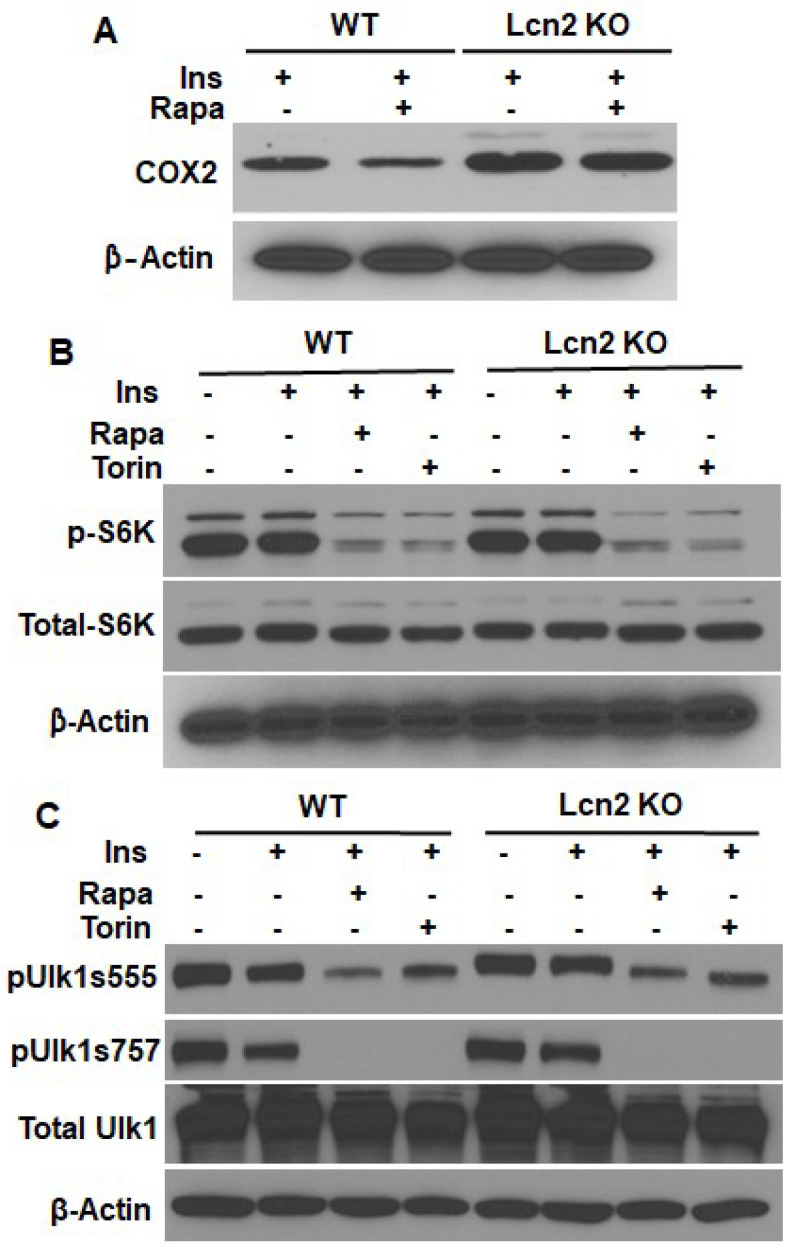
Effect of mTORC1 inhibition by rapamycin on COX2, p70 S6Kinase, and ULK1 activation in inguinal adipocytes. (**A**) COX2 protein levels by rapamycin in WT and Lcn2 KO adipocytes. Phosphorylation of p70 S6Kinase (**B**) and ULK1 (**C**) by rapamycin and Torin in the presence of insulin (Ins) in WT and Lcn2 KO adipocytes. Blots represent the results from one of 2–3 experiments.

**Figure 5 cells-11-01535-f005:**
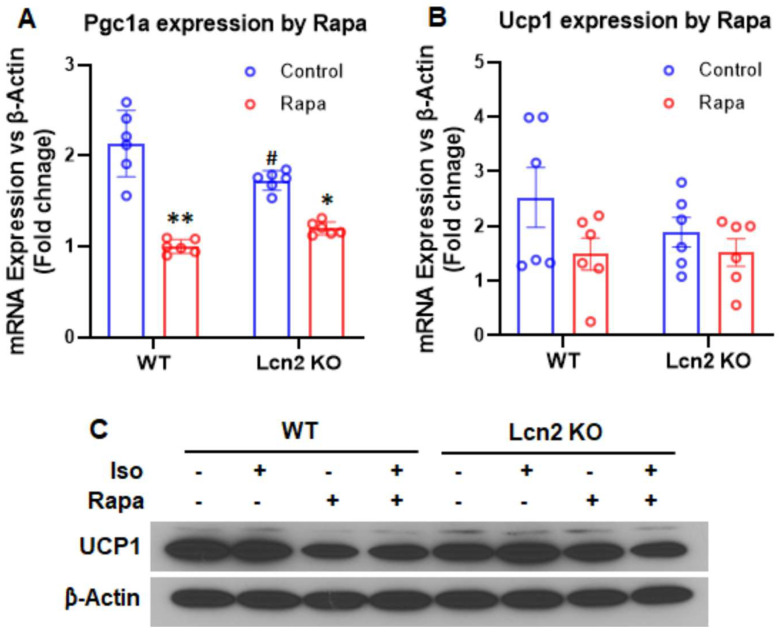
Effect of mTORC1 inhibition by rapamycin on thermogenic gene and protein expression in inguinal adipocytes. (**A**,**B**) The mRNA expression of *Pgc1α* and *Ucp1* genes by rapamycin in Lcn2 KO adipocytes, respectively. (**C**) The protein expression of UCP1 by rapamycin in Lcn2 KO adipocytes. Results are presented as mean ± SEM (*n* = 6 for (**A**,**B**)). * *p* < 0.05, ** *p* < 0.01 versus controls. ^#^
*p* < 0.05 versus WT adipocytes.

**Figure 6 cells-11-01535-f006:**
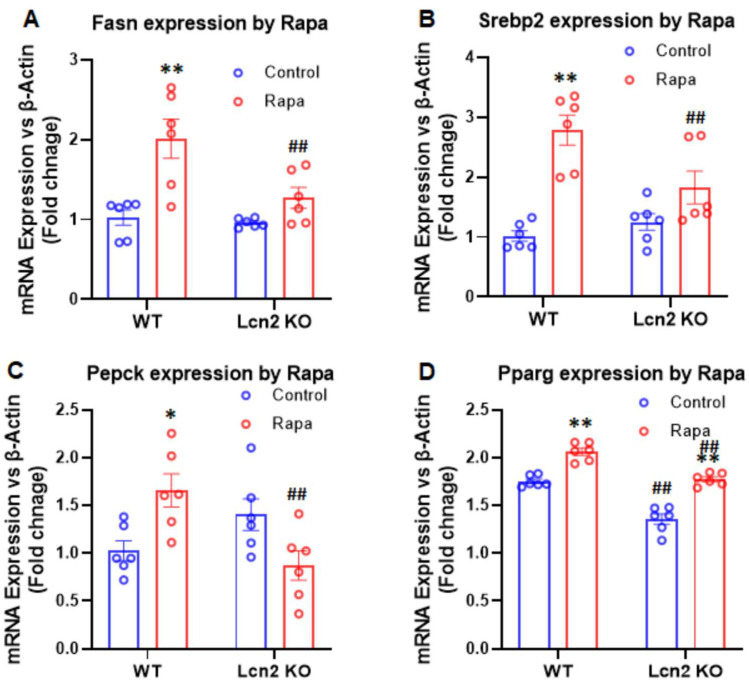
Effect of mTORC1 inhibition by rapamycin on lipogenesis gene expression in inguinal adipocytes (**A**–**D**). Results are presented as mean ± SEM (*n* = 6 for (**A**–**D**)). * *p* < 0.05, ** *p* < 0.01 versus controls. ^##^
*p* < 0.01 versus WT adipocytes.

**Figure 7 cells-11-01535-f007:**
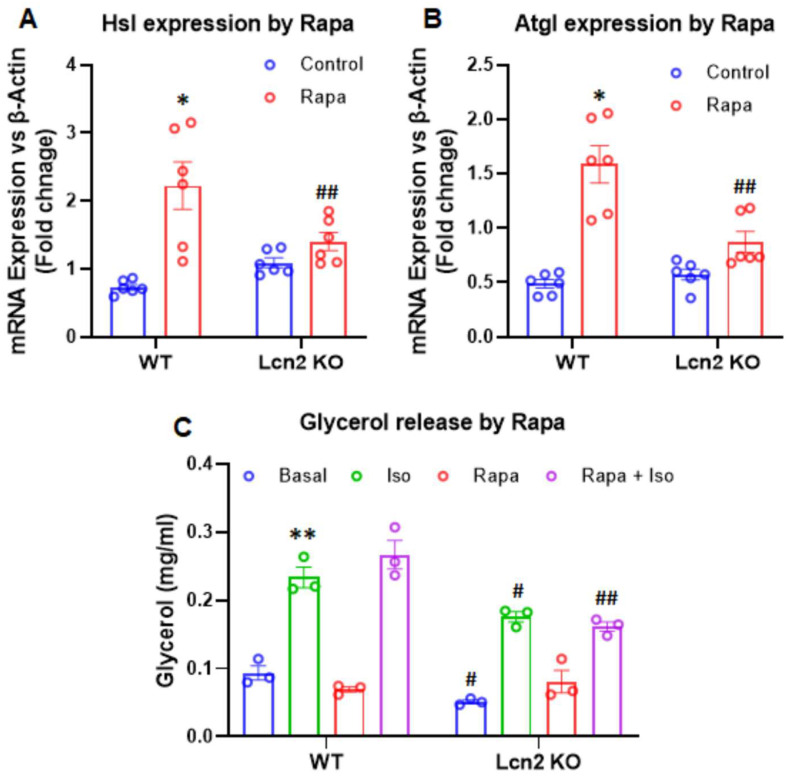
Effect of mTORC1 inhibition by rapamycin on lipolysis in inguinal adipocytes. (**A**,**B**) The mRNA expression of *Hsl* and *Atgl* genes by rapamycin in Lcn2 KO adipocytes. (**C**) Glycerol release in response to the stimulation of isopropanol (Iso), as well as the combination of Iso and rapamycin. Results are presented as mean ± SEM (*n* = 6 for (**A**,**B**); *n* = 3 for (**C**)). * *p* < 0.05, ** *p* < 0.01 versus controls. ^#^
*p* < 0.05, ^##^
*p* < 0.01 versus WT adipocytes. Panel C represents the results from one of two experiments.

## Data Availability

Not applicable.

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
