# Peer review of "Lipocalin 2 Deficiency Alters Prostaglandin Biosynthesis and mTOR Signaling Regulation of Thermogenesis and Lipid Metabolism in Adipocytes"

_cells, 2022, doi:10.3390/cells11091535_

Round 1

Reviewer 1 Report

The manuscript titled: “Lipocalin 2 Deficiency Alters Prostaglandin Biosynthesis and mTOR Signaling Regulation of Thermogenesis and Lipid Metabolism in Adipocytes” is well written. However, data needs to be refined in a presentable manner. The present manuscript would be benefited by addressing the points below.

Major comments:

  1. In figure 1, the authors have shown that arachidonic acid (AA) induces the Lcn2 expression and secretion through NF-kB signaling in adipocytes.  They have used the Lcn2 knockout cells to check the AA effect on the Lcn2 in adipocytes. I don't think it supports the hypothesis. I would suggest authors use NF-kb inhibitor in the AA treated adipocytes to check the expression/ secretion of Lcn2 through the NF-kB signaling.
  2. In figure 3, there is no description about section C in figure 3.

Author Response

Response to reviewers’ comments are also attached 

Reviewer 1

The manuscript titled: “Lipocalin 2 Deficiency Alters Prostaglandin Biosynthesis and mTOR Signaling Regulation of Thermogenesis and Lipid Metabolism in Adipocytes” is well written. However, data needs to be refined in a presentable manner. The present manuscript would be benefited by addressing the points below.

 Major comments:

  1. In figure 1, the authors have shown that arachidonic acid (AA) induces the Lcn2 expression and secretion through NF-kB signaling in adipocytes.  They have used the Lcn2 knockout cells to check the AA effect on the Lcn2 in adipocytes. I don't think it supports the hypothesis. I would suggest authors use NF-kb inhibitor in the AA treated adipocytes to check the expression/ secretion of Lcn2 through the NF-kB signaling.

 RESPONSE: We agree it would be better to show if the inhibition of NFkB would block AA-induced Lcn2 expression. The reason for why we did not include this is because we indeed have already done the similar experiment as the reviewer suggested and published the data showing the role of NFkB signaling in mediating the inflammation induction of Lcn2 expression and secretion (Ref# 10). For instance, blocking NFkB using either Asprin or Bay could significantly reduce LPS-induced Lcn2 expression and secretion. Additionally, studies from others have shown there is a NFkB binding site in the promoter region of Lcn2 gene. Thus, the literature and our previous studies have already established the relationship between NFkB and Lcn2 expression.  

  1. Zhang, Y.; Foncea, R.; Deis, J.A.; Guo, H.; Bernlohr, D.A.; Chen, X. Lipocalin 2 Expression and Secretion Is Highly Regulated by Metabolic Stress, Cytokines, and

In response to the reviewer’s comment, we have revised the statements regarding this in the discussion section as follows.

“It is likely that AA treatment induces Lcn2 expression via activating the NFkB inflammatory pathway. This possibility is supported by the fact that Lcn2 gene promoter region contains NFkB binding site, as well as our previously published data showing that blocking NFkB activation could significantly reduce inflammation-induced Lcn2 expression and secretion [10].”

        2. In figure 3, there is no description about section C in figure 3.

RESPONSE: Thanks for the comment. We have added the description of Fig 3C results as follows.

“Likewise, Lcn2 protein secretion was induced by rapamycin in a time- and dose-dependent manner (Figure 3B and 3C). While 6 hours of rapamycin treatment had no effect on Lcn2 secretion, 24 and 48 hours of treatment increased the secretion of Lcn2 protein (Figure 3B). Additionally, we showed the dose-dependent effect of rapamycin on Lcn2 secretion (Figure 3C). When treated for 24 hours, only 25nM but not 2.5nM rapamycin was able to stimulate the secretion of Lcn2. In both time- and dose-dependent studies, the effect of rapamycin requires the presence of insulin.  

Reviewer 2 Report

Lipocalin 2 (Ln2) is a hydrophobic ligand-binding protein, belonging to the lipocalin supergene family. In this manuscript, the authors examined how Lcn2 regulates thermogensis and lipid metabolism in adipose cells with special reference to mTOR signaling. There are some concerns as follows.

Major concerns:

  1. In Figure 1, is the induction of Lcn2 by arachidonic acid through the NFkB inflammatory pathway mediated by prostaglandins produced from arachidonic acid by COX-2?
  2. In Figure 4B, the reduction of the p70 S6 kinase phosphorylation by rapamycin alone is not clear.
  3. In Figure 4C, the reduction of the ULK1 phosphorylation by rapamycin alone is not clear.
  4. (Line 227) It is unclear whether “similar phenomenon” was observed in Figure 5B.
  5. In Figure 5C, semi-quantitative analysis should be necessary. The authors should also refer to Iso in Figure 5C.
  6. In Figure 6, PEPCK is an enzyme involved in gluconeogenesis rather than lipogenesis.

Minor concerns:

  1. 2 of prostaglandin E2, D2, J2 , and PLA2 should be subscript.
  2. (Line 208) siganling should be signaling.
  3. (Line 227) “Phenomena was” should be “Phenomena were”.

Author Response

Reviewer 2

Lipocalin 2 (Ln2) is a hydrophobic ligand-binding protein, belonging to the lipocalin supergene family. In this manuscript, the authors examined how Lcn2 regulates thermogensis and lipid metabolism in adipose cells with special reference to mTOR signaling. There are some concerns as follows.

 Major concerns:

  1. In Figure 1, is the induction of Lcn2 by arachidonic acid through the NFkB inflammatory pathway mediated by prostaglandins produced from arachidonic acid by COX-2?

 RESPONSE: We agree with the reviewer’s point. The following statement has been added to the manuscript in the Discussion section.

“Additionally, it is also possible that AA induces Lcn2 expression indirectly through stimulating PG production which in turn activates NFkB and Lcn2 expression.”

        2. In Figure 4B, the reduction of the p70 S6 kinase phosphorylation by rapamycin alone is not clear.

RESPONSE: We apologize for the mistake we made in the labeling of the treatments for the western blots in Fig 4B. This has been corrected. Thanks for pointing this out.

         3. In Figure 4C, the reduction of the ULK1 phosphorylation by rapamycin alone is not clear.

RESPONSE: Thanks for pointing this out. We made a similar mistake in Fig 4C as we did in Figure 4B. We have corrected the labeling of the treatments.

        4. (Line 227) It is unclear whether “similar phenomenon” was observed in Figure 5B.

RESPPONSE: We agree. This has been revised as indicated in the response to Question #5 below.

       5. In Figure 5C, semi-quantitative analysis should be necessary. The authors should also refer to Iso in Figure 5C.

RESPPONSE: Thanks for this point. We have quantified band intensities from two experiments. The changes did not reach a statistical significance level.  We therefore revised the description of the results for Figure 5B and 5C as follows.

“Although they did not reach a statistical significance level, the changes in Ucp1 gene (Figure 5B) and UCP1 protein expression (Figure 5C) followed a similar trend in WT and Lcn2 KO adipocytes treated with rapamycin for 24hrs.  Rapamycin treatment had less effect on the downregulation of Ucp1 gene expression in Lcn2 KO adipocytes compared to WT adipocytes (Figure 5B).  UCP1 protein expression was reduced by rapamycin treatment in the basal and Iso-treated conditions in WT adipocytes but not in Lcn2 KO adipocytes (Figure 5C). However, quantitative band intensities of western blotting results from two experiments indicate that the differences between control and rapamycin-treated groups did not reach a statistical significance level in either WT or Lcn2 KO adipocytes (data not shown).”         

        6. In Figure 6, PEPCK is an enzyme involved in gluconeogenesis rather than lipogenesis.

RESPONSE: In addition to the role in gluconeogenesis, phosphoenolpyruvate carboxykinase (PEPCK) is also a key regulator of glyceroneogenesis pathway which contributes to triglyceride synthesis via recycling fatty acids during the fasting. This pathway is particularly active in adipose tissue. Please see below for more information (from Wikipedia).   

“Glyceroneogenesis is a metabolic pathway which synthesizes glycerol 3-phosphate or triglyceride from precursors other than glucose.[1] Usually glycerol 3-phosphate is generated from glucose by glycolysis, but when glucose concentration drops in the cytosol, it is generated by another pathway called glyceroneogenesis. Glyceroneogenesis uses pyruvate, alanine, glutamine or any substances from the TCA cycle as precursors for glycerol 3-phosphate. Phosphoenolpyruvate carboxykinase (PEPC-K),[1] which is an enzyme that catalyzes the decarboxylation of oxaloacetate to phosphoenolpyruvate is the main regulator for this pathway. Glyceroneogenesis can be observed in adipose tissue and also liver.”

Minor concerns:

  1. 2 of prostaglandin E2, D2, J2 , and PLA2 should be subscript.

 RESPONSE: This has been fixed.

     2. (Line 208) siganling should be signaling.

 RESPONSE: This has been corrected.

     3. (Line 227) “Phenomena was” should be “Phenomena were”.

 RESPONSE: This has been changed.